
# High-frequency monitoring of anomalous methane point sources with multispectral Sentinel-2 satellite observations

Daniel J. Varon[1,2], Dylan Jervis[2], Jason McKeever[2], Ian Spence[2], David Gains[2], Daniel J. Jacob[1]

[1]School of Engineering and Applied Science, Harvard University, Cambridge, 02138, United States
[2]GHGSat, Inc., Montréal, H2W 1Y5, Canada

*Correspondence to*: Daniel J. Varon (danielvaron@g.harvard.edu)

**Abstract.** We demonstrate the capability of the Sentinel-2 MultiSpectral satellite Instrument (MSI) to detect and quantify large methane point sources with fine pixel resolution (20 m) and rapid revisit rates (2-5 days). We present three methane column retrieval methods that use shortwave infrared (SWIR) measurements from MSI spectral bands 11 (~1560-1660 nm) and 12
(~2090-2290 nm) to detect atmospheric methane plumes. The most successful is the multi-band/multi-pass (MBMP) method, which uses a combination of the two bands and a non-plume control observation to retrieve methane columns. The MBMP method can quantify point sources down to about 3 t h$^{-1}$ with precision of ~30%-90% (1$\sigma$) over favourable (quasi-homogeneous) surfaces. We applied our methods to perform high-frequency monitoring of strong methane point source plumes from a well-pad device in the Hassi Messaoud oil field of Algeria (October 2019 to August 2020, observed every 2.5 days)
and from a compressor station in the Korpezhe oil/gas field of Turkmenistan (August 2015 to November 2020, observed every 5 days). The Algerian source was detected in 93% of cloud-free scenes, with source rates ranging from 2.6 to 51.9 t h$^{-1}$ (averaging 9.3 t h$^{-1}$) until it was shut down by a flare lit in August 2020. The Turkmen source was detected in 40% of cloud-free scenes, with variable intermittency and a 9-month shutdown period in March-December 2019 before it resumed; source rates ranged from 3.5 to 92.9 t h$^{-1}$ (averaging 20.5 t h$^{-1}$). Our source rate retrievals for the Korpezhe point source are in close
agreement with GHGSat-D satellite observations for February 2018 to January 2019, but provide much higher observation density. Our methods can be readily applied to other satellite instruments with coarse SWIR spectral bands, such as Landsat-7 and Landsat-8. High-frequency satellite-based detection of anomalous methane point sources as demonstrated here could enable prompt corrective action to help reduce global methane emissions.

## 1 Introduction

Methane is a potent greenhouse gas that is responsible for roughly one quarter of the climate warming experienced since preindustrial times (IPCC, 2013). Natural methane emissions are primarily from wetlands. Anthropogenic emissions originate from a myriad of point sources associated primarily with livestock, coal mining, oil/gas production, and waste management (Saunois et al., 2020). Measurement surveys of methane-emitting facilities have shown that a small number of anomalously strong point sources contribute a large fraction of total emissions, due to equipment malfunction and/or abnormal operating





conditions (Frankenberg et al., 2016; Zavala-Araiza et al., 2017; Duren et al., 2019). This presents an opportunity for effective climate change mitigation if the strongest point sources can be rapidly identified, repaired, and routinely monitored.

Satellite observations of atmospheric methane by solar backscatter in the shortwave infrared (SWIR) have unique potential for global and individual monitoring of point sources, but a combination of fine spatial resolution and frequent revisit rate is needed. The TROPOspheric Monitoring Instrument (TROPOMI) aboard the Sentinel-5 Precursor satellite provides daily

global methane measurements at $5.5 \times 7$ km$^2$ pixel resolution (Hu et al., 2018), sufficient to detect major accidental blowouts at oil/gas facilities (Pandey et al., 2019), but generally too coarse to resolve point sources, which are often spatially clustered and typically produce plumes <1 km in scale (Frankenberg et al., 2016; Duren et al., 2019). GHGSat microsatellite instruments (Jervis et al., 2020; Ramier et al., 2020) are specifically designed to detect methane point sources using fine pixel resolution (25-50 m) over limited domains ($12 \times 12$ km$^2$) and with relatively high precision (~1%-15%). Hyperspectral imaging

spectrometers designed to observe land surfaces at 1-10 nm spectral resolution with 30-m pixel resolution can detect large methane plumes (Thompson et al., 2016; Cusworth et al., 2019), as was recently demonstrated with the Italian Space Agency's PRISMA instrument (Cusworth et al., 2020). Revisit times for these targeting instruments are limited by spatial coverage, tasking constraints, and the number of satellites; achieving frequent revisits will require a constellation.

Here we demonstrate the capability of the current Sentinel-2 twin satellites to detect and quantify strong methane

point sources globally with both fine pixel resolution and frequent revisits. Sentinel-2 comprises two satellites positioned 180° out of phase in the same sun-synchronous orbit, with an equator-crossing time of 10:30 (local solar time) at the descending node. Sentinel-2A (S2A) was launched in June 2015 and Sentinel-2B (S2B) in March 2017. Each satellite carries a MultiSpectral Instrument (MSI) that continuously sweeps the Earth's surface in 13 spectral bands from the visible to the shortwave infrared (SWIR), at 10-60 m pixel resolution over a 290-km cross-track swath (Drusch et al., 2012). The twin

satellite configuration enables full global coverage every 5 days and 2-3 day revisit rates at midlatitudes. We show here that Sentinel-2 SWIR bands 11 (~1560-1660 nm) and 12 (~2090-2290 nm), with 20-m pixel resolution, can be used to detect plumes from large methane point sources and quantify source rates. These bands integrate radiances over methane's 1650-nm and 2300-nm SWIR absorption features. Band 12, overlapping with the stronger and broader 2300-nm feature, is considerably more sensitive to methane than band 11. Band 11 can therefore be used as a proxy for the continuum, being spectrally close to

band 12 and having generally similar surface reflectances. Although the MSI spectral resolution of ~100-200 nm is far too coarse to permit standard retrieval of methane column concentrations by hyperspectral fitting in the SWIR, methane columns can still be derived from reflectance differences between the spectral bands and between satellite passes.

We present three different retrieval approaches that use Sentinel-2 data from bands 11 and 12 on one or more satellite passes to derive methane column enhancements across a scene. We assess retrieval error in each of these approaches for a

variety of scenes and surface types, and estimate associated plume detection limits. Furthermore, we present case studies illustrating high-frequency monitoring of methane emissions from venting at two oil/gas facilities in the Hassi Messaoud oil field of Algeria and the Korpezhe oil/gas field of Turkmenistan, inferring source rates for more than 160 methane plumes





observed by Sentinel-2 at these sites between 2015 and 2020. This offers a unique perspective on the variability and intermittence of methane emissions from large point sources.

Our techniques are developed with a focus on Sentinel-2 satellite observations, owing to the exceptional spatial and temporal resolution of MSI data, but can easily be extended to observations from other multispectral surface imagers with similar spectral bands, such as the Landsat 7 and Landsat 8 instruments with 30-m pixel resolution and a combined 8-day revisit rate. This work demonstrates how spaceborne multispectral imaging instruments can facilitate global high-frequency mapping of large methane point sources by combining fine pixel resolution with rapid revisit rates.

## 2 Sentinel-2 Data

We use Sentinel-2 level 1C (L1C) data for top-of-atmosphere reflectances in spectral bands 11 and 12 to retrieve methane column enhancements of individual plumes. The Sentinel-2 data are openly available on the European Space Agency's Copernicus Open Access Hub, and are provided at 20-m pixel resolution over $100 \times 100$ km$^2$ surface tiles with fixed geographic coordinates in UTM/WGS84 projection (Drusch et al., 2012; ESA 2020a). We present retrievals for two locations observed

between June 2015 and October 2020. The first location is a device at a well pad in the Hassi Messaoud oil field of Algeria (31.6585°N, 5.9053°E; tile 32SKA). The second is a device at a compressor station in the Korpezhe oil/gas field of Turkmenistan (38.4939°N, 54.1977°E; tile 40SBH), which was previously studied by Varon et al. (2019) using the GHGSat-D demonstration satellite instrument. Sentinel-2 acquired 109 cloud-free observations at the Algerian facility over a 10-month period and 171 at the Turkmen facility over a 5-year period (see Section 4).

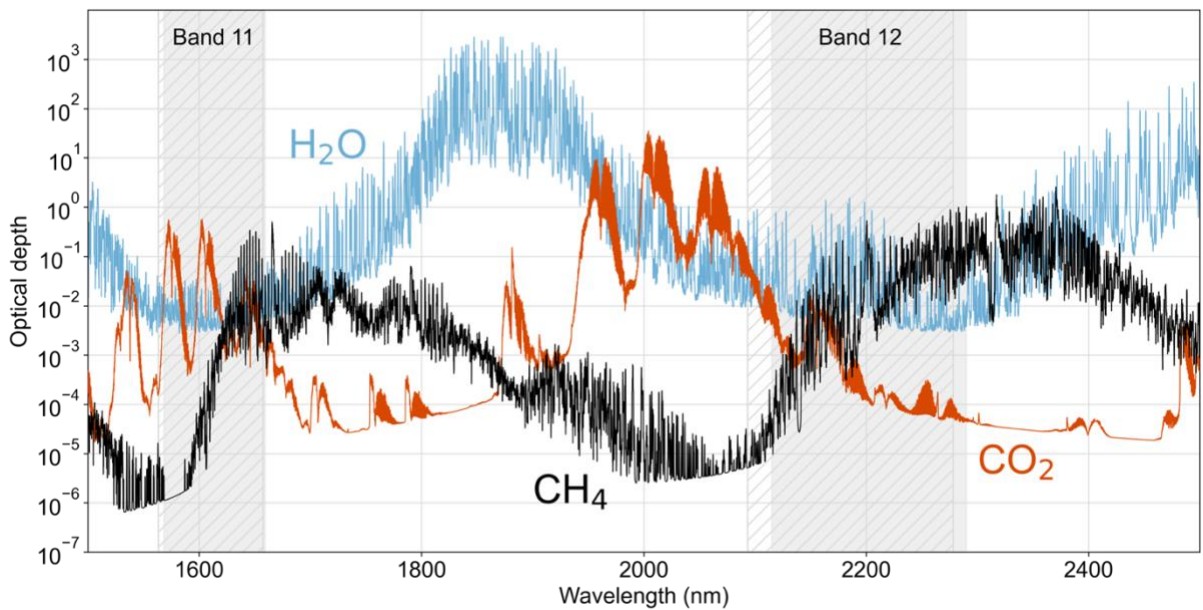


**Figure 1:** Methane (CH$_4$) , CO$_2$, and water vapor (H$_2$O) slant optical depths in the 1500-2500 nm SWIR spectral range, based on absorption line strengths from the HITRAN database sampled at 20-pm spectral resolution. Values are for the U.S. Standard Atmosphere (Anderson et


al., 1986), with surface concentrations adjusted to 1875 ppb for methane and 410 ppm for $CO_2$. The slant optical depth calculation is done for a solar zenith angle of 40° and satellite viewing angle of 0°. The optical depths are smoothed with a 20-point moving average for visual

clarity. The grey shaded areas are the spectral ranges of bands 11 and 12 for Sentinel-2A (solid) and Sentinel-2B (hatched).

Figure 1 shows the spectral bandwidths of Sentinel-2 bands 11 and 12 along with methane, $CO_2$, and water vapor optical depths in the 1500-2500 nm SWIR spectral range, based on absorption line spectra from the high-resolution transmission molecular absorption (HITRAN) database (Gordon et al., 2017). The MSIs aboard Sentinel-2A and Sentinel-2B have slightly different spectral transmission window positions and widths (Figure 1). Bands 11 and 12 for Sentinel-2A cover

1568.2-1659.2 nm and 2114.9-2289.9 nm, respectively; for Sentinel-2B, the ranges are 1563.4-1657.4 and 2093.2-2278.2 (ESA, 2020b). Band 11 extends over a set of weak methane absorption lines near 1650 nm. Band 12 includes stronger absorption lines over the 2200-2300 nm range. The mean methane optical depth in band 12 is five times greater than that in band 11. Water vapor absorption lines in bands 11 and 12 introduce a risk of retrieval artefacts, but we show in Section 3.1 that the effect is relatively small.

## 95 3 Methane column retrievals

Standard retrieval algorithms estimate vertical column concentrations [mol m$^{-2}$] of atmospheric methane by fitting a radiative transfer model to remotely-sensed SWIR spectra. Typically the spectra are highly resolved, with full width at half maximum of 0.1 to 10 nm and tens to thousands of spectral samples (Hamazaki et al., 2005; Jacob et al., 2016; Cusworth et al., 2019; Hasekamp et al., 2019). This enables joint optimization of methane, other trace gases, and surface albedo from a single

observation. But one can also in principle retrieve methane column concentrations together with surface albedo from just two spectral measurements, one featuring methane absorption and one not. This could be done for a single spectral band by comparing observations of the same scene with and without a methane plume. It could also be done for a single scene with two adjacent spectral bands that are sufficiently close to have similar surface and aerosol reflectance properties, but differ in their methane absorption properties. We demonstrate this here using Sentinel-2 bands 11 and 12.

We use a 100-layer, clear-sky radiative transfer model to simulate top-of-atmosphere (TOA) radiances in the Sentinel-2 SWIR bands (Figure 1) for comparison with Sentinel-2 measurements. The model accounts for molecular absorption but not thermal emission or scattering (Jervis et al., 2020). We simulate radiances at 0.02 nm spectral resolution and integrate them over the band-11 and band-12 spectral windows. The model relies on U.S. Standard Atmosphere vertical profiles of pressure, temperature, air density, water vapor, $CO_2$ (scaled to 410 ppm at sea level), and background methane (scaled to 1875 ppb at

sea level, or 0.65 mol m$^{-2}$ in the column). Absorption line strengths for the trace gases are obtained from the HITRAN database and convolved with a Voigt profile to compute absorption cross-sections (Kochanov et al., 2016). The incident solar irradiance is from Clough et al. (2005). The radiative transfer model accounts for variable surface height, solar zenith angle (SZA), and instrument viewing zenith angle (VZA). We obtain SZA and VZA for each scene from metadata provided with the Sentinel-2 image tiles, and surface height at the source location from © Google Earth elevation data. Aerosol effects are ignored because





we assume that they would be (1) uniform across the scene, (2) relatively small because aerosol is generally not co-emitted
with methane, and (3) partly corrected when using both bands 11 and 12 to retrieve methane concentrations.

Our Sentinel-2 retrievals employ a general strategy of deriving methane column enhancements $\Delta\Omega$ [mol m$^{-2}$] from
fractional changes in TOA reflectances for band 12 relative to a "control" image without (or with less) SWIR absorption by a
methane plume. Examining fractional changes means that methane column concentrations can be retrieved from any
radiometric measurement that is proportional to TOA radiance, including reflectances and even 8-bit image brightness data.
Here we use TOA reflectances from the Sentinel-2 L1C product.

**3.1 Single-band/multi-pass (SBMP) retrieval**

The first of our three column retrieval methods is a single-band method that compares band-12 TOA reflectances $R_{12}$ measured
over an active methane point source with values $R'_{12}$ measured over the same location on a day without emissions. This
presumes some intermittence in emissions, which is characteristic of many methane point sources (Duren et al., 2019). We
derive methane column concentrations from the fractional change in reflectance:

$$\Delta R_{SBMP} = \frac{cR_{12} - R'_{12}}{R'_{12}}, \tag{1}$$

where $c$ is a scaling factor that adjusts for scene-wide changes in brightness between satellite passes. We calculate $c$ by least-
squares fitting of all values of $R_{12}$ in the scene against all values of $R'_{12}$, using a first-order linear regression with an intercept
of zero. To infer the methane column enhancement $\Delta\Omega$, we compare $\Delta R_{SBMP}$ to a fractional absorption model:

$$m_{SBMP}(\Delta\Omega) = \frac{T_{12}(\Omega + \Delta\Omega) - T_{12}(\Omega)}{T_{12}(\Omega)}, \tag{2}$$

where $T_{12}(\Omega)$ is the simulated TOA spectral radiance for the nominal methane column concentration $\Omega$ [mol m$^{-2}$], integrated
over the band-12 spectral range and including absorption by $CO_2$ and $H_2O$. The methane enhancement is presumed to be in
the lowest 500 m of the atmosphere. We use a Gauss-Newton method to retrieve the enhancement by minimizing $F(\Delta\Omega) =$
$\Delta R_{SBMP} - m_{SBMP}(\Delta\Omega)$. For a scene at sea level, with the satellite positioned directly overhead and the Sun at 40° from nadir
(VZA = 0°, SZA = 40°), a doubling of the methane column yields $m_{SBMP} = -0.036$ for Sentinel-2A and $-0.028$ for Sentinel-
2B, corresponding to fractional signal changes of 3.6% and 2.8%, respectively.

The SBMP retrieval has the benefit of conceptual simplicity but the disadvantage of requiring measurements from
more than one satellite pass. It may be challenging to identify a plume-free satellite pass when monitoring persistent methane
sources. Furthermore, the retrieval is vulnerable to non-uniform changes in surface albedo over time. We verified that changes
in background water vapour concentration between scenes would have only a small effect. Water vapor columns over land
may vary from 1 to 40 kg m$^{-2}$ (Nelson et al., 2016) but this affects $m_{SBMP}$ by only 6% for nominal observing conditions.



### 3.2 Multi-band/single-pass (MBSP) retrieval

Our second method is a multi-band retrieval that estimates methane enhancements from differences between the band-11 and
band-12 reflectances measured on a single satellite pass. For this method, the fractional change in reflectance is given by:

$$\Delta R_{\text{MBSP}} = \frac{cR_{12} - R_{11}}{R_{11}}, \tag{3}$$

where $c$ is now determined by least-squares fitting of $R_{12}$ against $R_{11}$ across the scene. The fractional absorption model is then:

$$m_{\text{MBSP}}(\Delta\Omega) = \frac{T_{12}(\Omega + \Delta\Omega) - T_{12}(\Omega)}{T_{12}(\Omega)} - \frac{T_{11}(\Omega + \Delta\Omega) - T_{11}(\Omega)}{T_{11}(\Omega)}. \tag{4}$$

This approach treats band 12 as an absorption band and band 11 as the continuum, relying on surface reflectance similarities
between the two adjacent bands. The empirical scaling factor $c$ now accounts for uncalibrated differences in signal throughput
between bands 11 and 12, plus spectral dependences of the surface albedo. The fractional absorption model accounts for the
non-zero methane sensitivity of band 11 by subtracting its simulated fractional absorption from that of band 12. We retrieve
$\Delta\Omega$ as in the SBMP method, this time by minimizing $F(\Delta\Omega) = \Delta R_{\text{MBSP}} - m_{\text{MBSP}}(\Delta\Omega)$. For a scene at sea level, with the
satellite positioned directly overhead and the Sun at 40° from nadir (VZA = 0°, SZA = 40°), we obtain $m_{\text{MBSP}}(\Omega_{CH_4}) =$
$-0.028$ for Sentinel-2A and $-0.023$ for Sentinel-2B.

The MBSP retrieval has the advantage of requiring just one satellite pass to retrieve methane concentrations, but the
disadvantage of using signals acquired from different spectral bands with central wavelengths separated by 600 nm.

### 3.3 Multi-band/multi-pass (MBMP) retrieval

Our third retrieval approach combines the techniques of the first two, deriving methane column enhancements from the
difference between MBSP retrievals from different satellite passes. In this method, we correct for systematic errors in the
MBSP retrieval $\Delta\Omega_{\text{MBSP}}$ due to wavelength separation between bands by subtracting another MBSP retrieval
$\Delta\Omega'_{\text{MBSP}}$ performed for a satellite pass when no methane plume was present:

$$\Delta\Omega_{\text{MBMP}} = \Delta\Omega_{\text{MBSP}} - \Delta\Omega'_{\text{MBSP}}. \tag{5}$$

If the systematic errors in the MBSP retrievals are similar on both passes, then this subtraction removes artefacts present in the
retrieval field, leaving only true methane enhancements.

### 3.4 Demonstration

Figure 2 shows examples of each of our three retrievals for a strong methane point source detected by Sentinel-2A at a well
pad in the Algerian Hassi Messaoud oil field on 20 November 2019 (see Section 4), along with band-11 and band-12 reflectance
data for that day and a previous "control" day when no plume was present (6 October 2019). We focus our attention on a 4×4
km$^2$ region around the point source. Methane emissions are not detectable in the band-12 reflectances on the control day (panel





2a), but band-12 extinction is evident on 20 November 2019, emanating northward from a device in the centre of the domain (panel 2b). No such extinction is apparent in the band-11 reflectances on 20 November 2019 (panel 2c), due to the weaker methane absorption lines in that spectral band (see Figure 1).

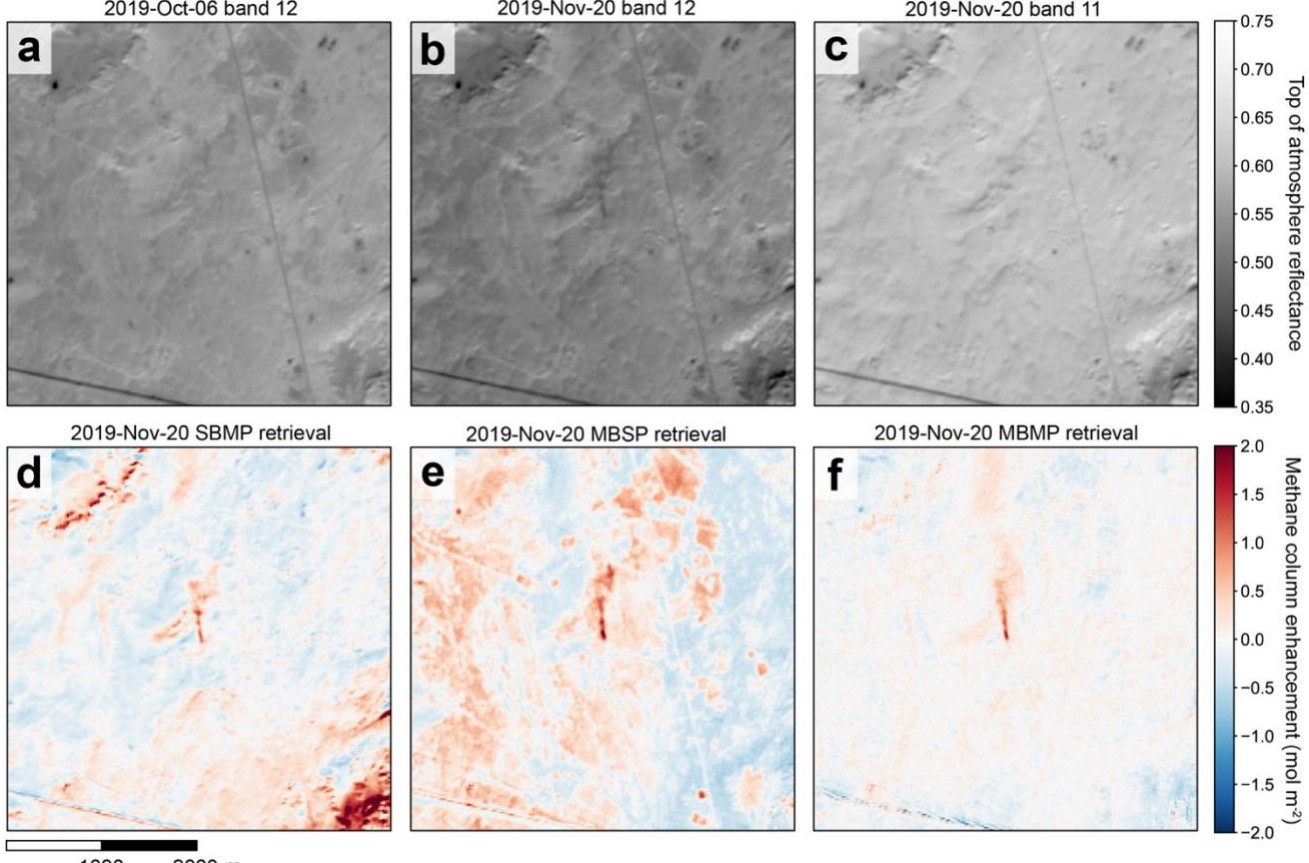

**Figure 2:** Demonstration of Sentinel-2 methane column retrieval for a plume emitted from a point source (31.6585°N, 5.9053°E) in the Hassi Messaoud oil field of Algeria, on 20 November 2019 (see Section 4). **(a)** Sentinel-2 top of atmosphere reflectances for band-12 observed on 6 October 2019. **(b)** The same as panel (a), but for 20 November 2019. **(c)** Top of atmosphere band-11 reflectances corresponding to panel (b). **(d)** Single-band/multi-pass (SBMP) retrieval for 20 November 2019. **(e)** Multi-band/single-pass (MBSP) retrieval for 20 November 2019. **(f)** Multi-band/multi-pass (MBMP) retrieval for 20 November 2019.

Panels 2d-f show the results of our three retrievals, with large methane plumes extending northward from the vent, but strong retrieval artefacts present in the domain. Estimating single-pixel column retrieval precision as the standard deviation of non-plume methane enhancements across the scene, we obtain precisions of 0.32 mol m$^{-2}$ (roughly 49% of background) for the SBMP retrieval and 0.31 mol m$^{-2}$ (roughly 48% of background) for the MBSP retrieval. The MBMP retrieval (panel 2f) produces by far the clearest methane plume, with considerably finer precision (0.13 mol m$^{-2}$, or 21% of background) and a much longer detectable plume extent than either of the other retrievals. We estimate for this plume a source rate of 8.5 ± 5.7 t h$^{-1}$ (see Section 4). Sentinel-2 single-pixel precision levels of 21%-49% are coarser than observed with GHGSat-D (9%-18%; Varon et al., 2020) but still permit quantification of strong methane point sources.





One might expect the MBMP retrieval to be strictly superior to the SBMP and MBSP retrievals since it exploits both multi-band and multi-pass information to derive methane column concentrations. However, this may not be the case for scenes

with large differences in surface reflectance between bands 11 and 12 due to strong spectral dependence of the albedo (in which case SBMP might be superior), or if no good control observation with consistent surface reflectance is available (in which case MBSP might be superior). We investigate the dependence of the methane retrieval precision on surface type and retrieval method in the following section.

**3.5 Retrieval precision and dependence on surface type**

Our Sentinel-2 methane column retrievals require close agreement between MSI surface reflectances measured on different satellite passes and/or in different SWIR spectral bands to isolate methane plumes in a scene. The scaling factor $c$ of Equations (1) and (3) is fitted to account for scene-averaged differences in reflectance between satellite passes or bands, but inter-pixel variability in surface reflectance remains and will be a source of error.

We illustrate the error from variable surface conditions for five different types of scenes: Hassi Messaoud , Korpezhe,

savannah, farmland, and urban. For each of these scenes but the last, we retrieve methane columns over a 4×4 km$^2$ area. We limit the urban scene to 2.4×2.4 km$^2$ to exclude non-urban areas from the retrieval domain. The Hassi Messaoud and Korpezhe scenes are masked to remove the methane plumes. The other three scenes have no apparent methane sources.

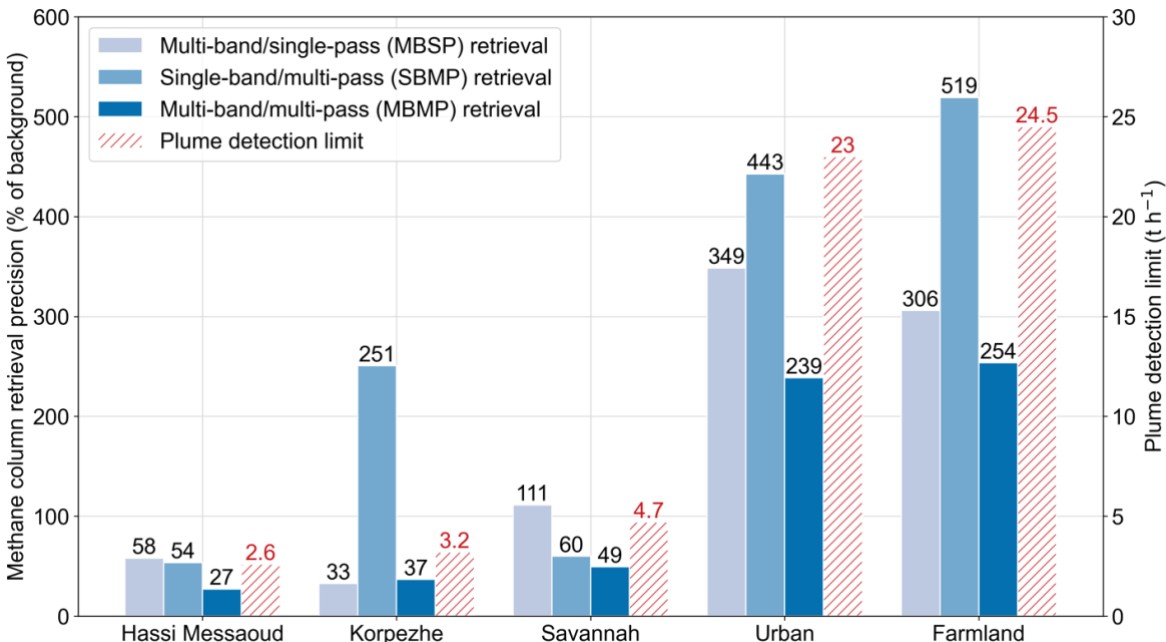

**Figure 3:** Single-pixel methane column retrieval precision as percentage of background for different scenes and retrieval methods (left
axis), and estimated lower limit for plume detection (right axis). The Hassi Messaoud and Korpezhe precisions are averages over all 68-101 satellite passes for those scenes (see Section 4). Detection limits are estimated empirically for Hassi Messaoud and Korpezhe (see Section 4); the savannah, urban, and farmland values are obtained by linear scaling of the Hassi Messaoud value for MBMP precision.



We use one control observation per Sentinel-2 MSI for all multi-pass (SBMP and MBMP) column retrievals performed at each site. The control observations for Hassi Messaoud (centred on 31.6585°N, 5.9053°E) are from 4 October

2019 (Sentinel-2A) and 6 October 2019 (Sentinel-2B); for Korpezhe (centred on 38.4939°N, 54.1977°E) they are from 2 August 2020 (Sentinel-2A) and 28 July 2020 (Sentinel-2B). The savannah scene is in the Mbeya Region of Tanzania (centred on 7.5778°S, 34.0918°E); we examine imagery from 20 and 30 July 2020 for this location. The urban scene is in Brooklyn, New York City (centred on 40.6113°N, 73.9952°W) and is observed on 6 and 19 July 2020. The farmland scene is in Saskatchewan, Canada (centred on 51.9908°N, 107.3488°W), with imagery from 24 and 31 July 2020. Mean band-11 (band-

12) reflectances for the Hassi Messaoud, Korpezhe, savannah, urban, and farmland scenes are 0.61, 0.35, 0.20, 0.26, and 0.15 (0.53, 0.32, 0.12, 0.21, and 0.06), respectively.

Figure 3 shows estimated single-pixel precisions for our three methane column retrievals over the five scenes, along with estimated plume detection limits. Precision is calculated as in Section 3.4, as the standard deviation of non-plume methane enhancements across the scene relative to a background column of 0.65 mol m$^{-2}$ (roughly 1875 ppb of methane). For the Hassi

Messaoud and Korpezhe retrievals, the reported precisions are averages over all 68-101 satellite passes. For the savannah, urban, and farmland scenes, the precision estimates reflect a single scene-wide column retrieval, and are intended only to be illustrative.

Precision is best using the MBMP retrieval for all scenes except Korpezhe, where the MBSP retrieval is slightly better. Precision is 27% for Hassi Messaoud and 33% for Korpezhe. Both of these scenes feature relatively homogeneous arid

surfaces. The more heterogeneous savannah scene has a precision of 49%, and the highly heterogeneous urban and farmland scenes have precisions in excess of 200%. These precisions could be improved by image segmentation to isolate different surfaces within the scene. For example, examining a relatively uniform 800×800 m$^2$ sub-area of the farmland scene (a single farm plot), we find a finer MBMP precision of 101%.

The dependence of retrieval precision on surface heterogeneity can be expressed explicitly. From equations (1) and

(3), it is evident that differences between $cR_{12}$ and $R'_{12}$ (or $R_{11}$) in image pixels without a true methane plume enhancement will produce errors in the retrieval field. Specifically, for small deviations $\delta\Omega$ from the background methane column, the standard deviation of the methane enhancement field, $\Delta\Omega^{SD}$, is related to the standard deviation $\Delta R^{SD}$ of the fractional reflectance field via:

$$\Delta\Omega^{SD} = \Delta R_x^{SD} \frac{\delta\Omega}{m_x(\delta\Omega)}, \tag{6}$$

where $x$ is either SBMP or MBSP.

The single-pixel column retrieval precisions of Figure 3 can be related to empirical point source detection limits by using the long-term observation records for the Hassi Messaoud and Korpezhe scenes shown in Section 4. For Hassi Messaoud with 27% retrieval precision, the minimum detected source is 2.6 t h$^{-1}$. For Korpezhe with 33% retrieval precision but more temporally variable surface conditions, the minimum detected source is 3.5 t h$^{-1}$. The plume detection limit is expected to be

proportional to precision (Jacob et al., 2016) and is shown in Figure 3 for the savannah, urban, and farmland scenes by scaling





of the Hassi Messaoud MBMP value. The largest reporting methane point sources emit 1-10 t h$^{-1}$ as annual means under normal operating conditions (Jacob et al., 2016; Varon et al., 2020; Scarpelli et al., 2020a). Our detection-limit estimates for Sentinel-2 thus restrict application to unusually strong sources, but such sources are frequently detected in abnormal operations (Pandey et al., 2019; Varon et al., 2019; Cusworth et al., 2020).

## 245 4 Application to high-frequency point-source monitoring

We apply our Sentinel-2 retrievals to high-frequency monitoring of two methane point sources, in the Hassi Messaoud oil field of Algeria and the Korpezhe oil/gas field of Turkmenistan. The Hassi Messaoud point source was discovered by inspection of Sentinel-2 imagery. GHGSat discovered the Korpezhe point source in January 2019, and TROPOMI corroborated its magnitude (Varon et al. 2019). Locations are shown in Figure 4. The Algerian point source is a piece of equipment near an oil

well pad. Sentinel-2 imagery indicates that it started emitting large quantities of methane on 9 October 2019 and continued until 9 August 2020, when a flare was lit (visible by Sentinel-2) and the plume became undetectable. The Turkmen point source is a piece of equipment about 400 m southwest of the Korpezhe compressor station. GHGSat-D observed plumes from this source beginning in June 2018, and TROPOMI detected plumes going back to November 2017 (Varon et al., 2019). Sentinel-2 first detected emissions from this source on 29 August 2015, soon after the launch of Sentinel-2A, and observed routine but

intermittent emissions up until at least 11 October 2020, the last observation day considered here.

**Hassi Messaoud oil field, Algeria**
(31.6585°N, 5.9053°E)

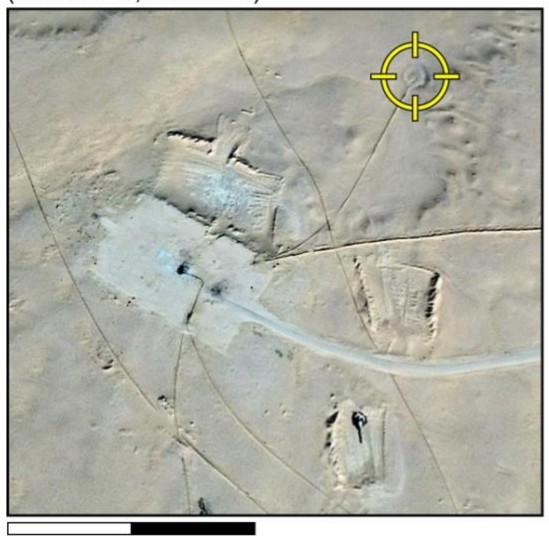

**Korpezhe oil/gas field, Turkmenistan**
(38.4939°N, 54.1977°E)

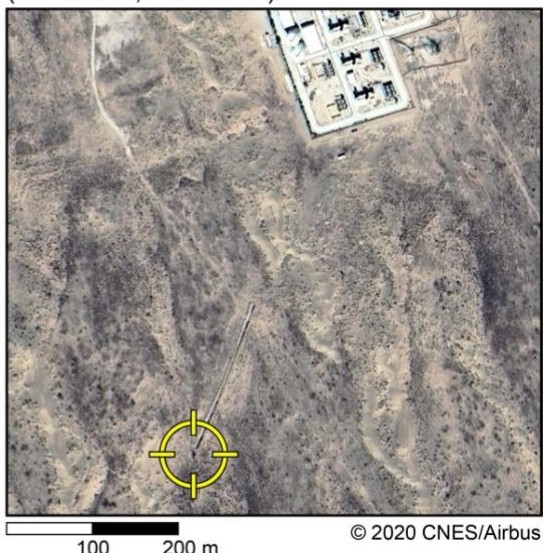

**Figure 4:** Surface imagery from © Google Earth showing the two oil/gas methane point sources to which our retrieval methods are applied for high-frequency Sentinel-2 monitoring. The point sources are pieces of equipment at the centres of the yellow target symbols. **(Left)** Equipment (31.6585°N, 5.9053°E) near an oil well pad in the Hassi Messaoud oil field of Algeria. The image is from 1 December 2018.
**(Right)** Equipment (38.4939°N, 54.1977°E) near a compressor station in the Korpezhe oil/gas field of Turkmenistan. The image is from 24 August 2019.



## 4.1 Source rate retrieval

We estimate source rates for each of our retrieved methane plumes using the integrated methane enhancement (IME) method (Frankenberg et al., 2016; Varon et al., 2018; Duren et al., 2019; Jongaramrungruang et al., 2019). As implemented by Varon et al. (2018), the IME method derives the source rate $Q$ from the total plume IME [kg], an effective wind speed $U_{\text{eff}}$ [m s$^{-1}$], and a plume length scale $L$ [m]:

$$Q = \frac{\text{IME} \times U_{\text{eff}}}{L}. \tag{7}$$

The IME is computed over a binary mask that separates the plume from the background; $L$ is the square-root of the area of the plume mask; and $U_{\text{eff}}$ is a function of the local 10-m wind speed $U_{10}$ that is calibrated with large-eddy simulations (LES) mimicking the satellite instrument specifications and the plume masking procedure. Here we define the plume mask by selecting methane columns above the 95$^{\text{th}}$ percentile for the scene and smooth with a 3×3 median filter.

Our LES ensemble used to calibrate the Sentinel-2 retrievals simulates plume transport at 25-m horizontal and 15-m vertical resolution over a 9×9×2.4 km$^3$ domain. The ensemble comprises five three-hour simulations with a range of sensible heat fluxes (100-300 W m$^{-2}$) and mixed layer depths (500-2000 m). We treat the first hour of each simulation as spin-up and use the remaining two hours for the calibration. After applying the Sentinel-2 single-pixel retrieval precision and plume mask, we find the following relationship between $U_{\text{eff}}$ and $U_{10}$:

$$U_{\text{eff}} = \alpha \, U_{10} + \beta, \tag{8}$$

where $U_{10}$ is an hourly mean, $\alpha = 0.33$, and $\beta = 0.45$ m s$^{-1}$. Since we lack local $U_{10}$ observations, we use 1-h average $U_{10}$ data from the NASA GEOS-FP meteorological reanalysis product at 0.25°×03125° resolution (Molod et al., 2012).

We estimate 1σ uncertainties for each of our reported source rates using the approaches of Varon et al. 2019, including contributions from wind speed error, retrieval error, and error in the IME model. Wind speed error is estimated by comparing GEOS-FP 10-m wind speed data with United States mesonet airport measurements from the MesoWest database (Horel et al. 2002). Retrieval error on the scale of the plumes is estimated from the variability in IME across the methane retrieval domain, by sampling the retrieved column enhancements with the plume mask at different background locations in the scene. Error in the IME model is evaluated with a test set of LES plumes not included in the effective wind speed calibration. For source rate estimates based on the multi-pass (SBMP and MBMP) methane column retrievals, we also include an uncertainty term reflecting sensitivity of the retrieved source rate to the control observation used in the column retrieval. This error term is computed as the RMS difference in source rates obtained using different pairs of control observations to retrieve methane columns (one observation for each of Sentinel-2A and Sentinel-2B). Standard error in fitting the scaling factor $c$ of Equations (1) and (3) is <1% for both the Hassi Messaoud and Korpezhe scenes. Total uncertainty is computed by quadrature addition of the individual error terms. We find total 1σ source-rate uncertainties ranging from 33%-86% for the Hassi Messaoud oil field point source and 21%-68% for the Korpezhe oil/gas field point source, generally dominated by wind speed error.



## 4.2 Hassi Messaoud oil field, Algeria

We identify 101 detectable methane plumes from the Hassi Messaoud point source out of 121 total Sentinel-2 satellite passes
between 9 October 2019 and 9 August 2020, corresponding to a pass on average every 2.5 days. Of the 20 non-detections, 12
were due to cloud cover and 8 showed no detectable plume, indicating a plume persistence rate of 93% for cloud-free
observations. Plume detection was determined for each cloud-free scene by using the SBMP, MBSP, and MBMP retrievals
across a 4×4 km$^2$ domain centred on the point source. We use control observations from 4 October 2019 (Sentinel-2A) and 6
October 2019 (Sentinel-2B), before the emissions began, for the multi-pass (SBMP and MBMP) retrievals. To assess
sensitivity to the control scene, we also perform multi-pass retrievals using control data from satellite passes that occurred
after the emissions ceased: 18 September 2020 (Sentinel-2A) and 20 September 2020 (Sentinel-2B).

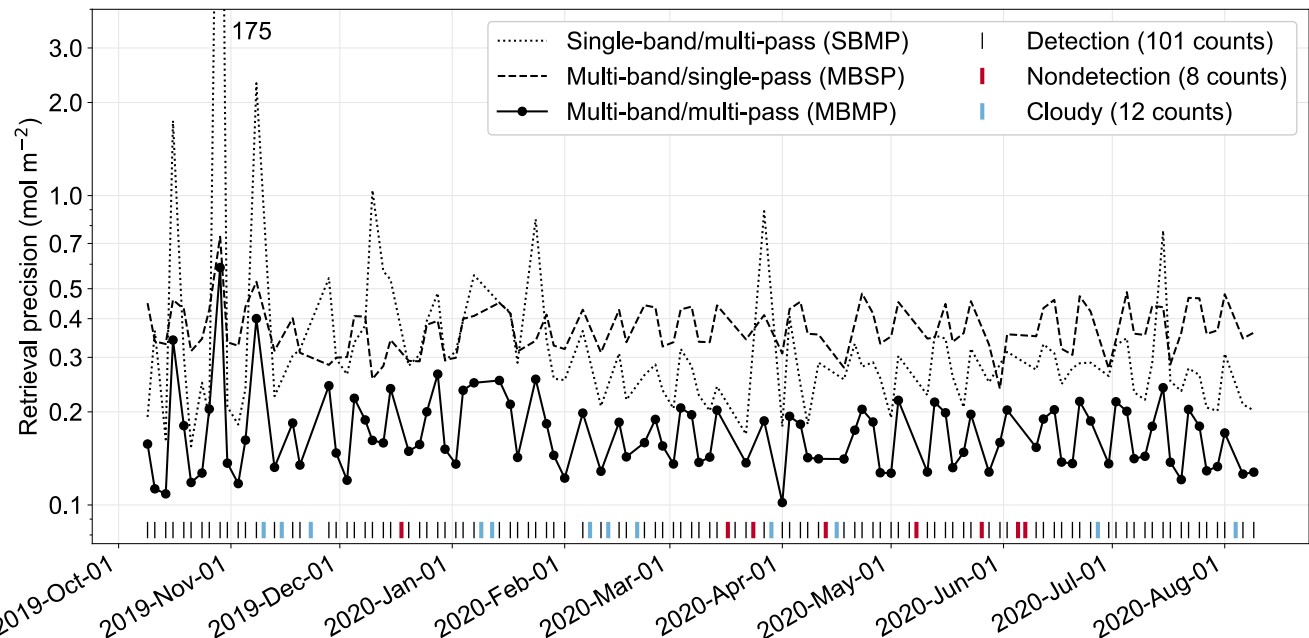

**Figure 5:** Single-pixel retrieval precision for the Hassi Messaoud scene over time, evaluated from the standard deviation of non-plume
enhancements retrieved across the scene. The black, red, and blue markers indicate which satellite passes had plume detections, which had
non-detections, and which were obscured by clouds, respectively. The y-axis is logarithmic.

Figure 5 shows the single-pixel column retrieval precision on days when a plume was detected, again estimated as
the standard deviation of non-plume methane enhancements across the scene. The MBMP retrieval shows consistently better
precision than the other two retrievals, as previously shown in Figure 2 and Figure 3. Precision fluctuates between about 0.1
and 0.25 mol m$^{-2}$, with a mean of 0.18 mol m$^{-2}$ (27% of background, value reported in Figure 3) and three outliers higher than
0.3 mol m$^{-2}$. These outliers are caused by partial cloud cover or large-scale gradients across the scene, which could be weather-
related. The consistent precision in the time series indicates that the value of our control observation for this scene does not
degrade over time. The SBMP and MBSP retrievals show much coarser precision, with mean (median) scene-wide standard



deviations of 0.35 (0.28) and 0.38 (0.36) mol m$^{-2}$, respectively (omitting the extreme outlier in October 2019 from the SBMP calculation). The MBSP shows the most consistent (but coarse) precision across the time series, presumably because it is not
subject to variability in surface conditions.

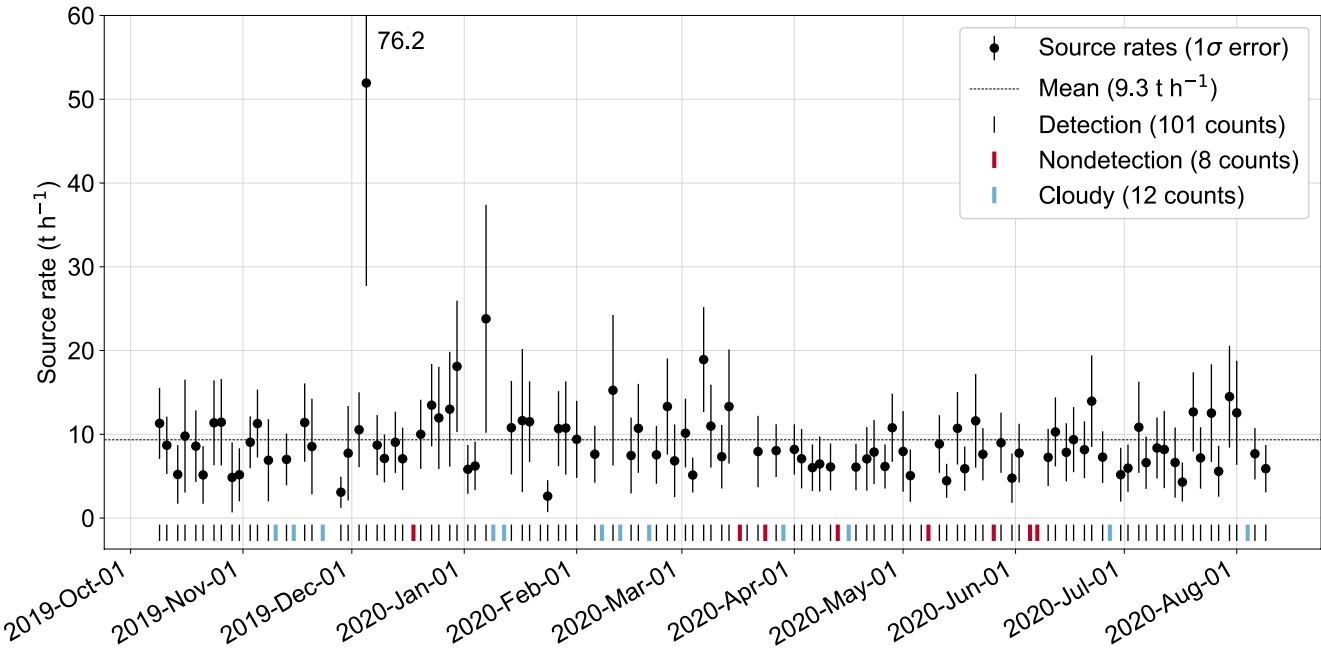

**Figure 6:** Time series of retrieved source rates for the Hassi Messaoud oil field point source, using the MBMP retrieval and the IME method.

Given its superior performance for this scene in the Hassi Messaoud oil field, we use the MBMP retrieval to construct methane plume masks and estimate source rates as described in Section 4.1. We estimate emission rates for 98 of the 101
plume detections, neglecting 3 detections for which retrieval artefacts were difficult to separate from the methane plume (these do not correspond to the 3 low-precision retrievals from October and November 2019 shown in Figure 5). Figure 6 shows the resulting time series of source rates, which range from 2.6 to 59.1 t h$^{-1}$, with a mean ± standard deviation of 9.3 ± 5.5 t h$^{-1}$. Using our second set of control observations, we obtain RMS differences of 20% between individual source rate retrievals as compared to the first set, and a mean source rate of 10.0 ± 6.0 t h$^{-1}$. This 20% error is included in the uncertainty estimates
quoted here and plotted in Figure 6 (see Section 4.1). Assuming a 93% persistence rate and no diurnal variability, the average of our two estimates of the mean source rate implies total methane emissions of 66 Gg over the course of the 10-month-long emission event. This represents 6% of the annual national methane emission of 1.19 Tg from the oil/gas sector reported by the Algerian government to the United Nations Framework Convention on Climate Change (UNFCCC; Scarpelli et al., 2020b). Given the facility operators' ultimate intervention by flaring in August 2020, it would appear that a large proportion of these
emissions could have been avoided had the operators been alerted soon after the emissions began. Sentinel-2 imagery can enable such intervention in the future.





### 4.3 Korpezhe oil/gas field, Turkmenistan

The Korpezhe source was intermittent over the full Sentinel-2 observational record from 9 August 2015 to 26 October 2020. Of the 291 satellite passes over the scene, 68 had detectable plumes, 120 were cloudy, 101 had no detectable plume, and 2 had

missing data records. The persistence rate for the Korpezhe compressor station source is thus 40% for non-cloudy observations. During the 5-year measurement period, Sentinel-2 observed Korpezhe once every 6.5 days on average. After summer 2017, when both satellites were fully operational, the revisit rate increased to one observation every 5 days. For each satellite pass with a detectable plume, we again perform our three methane column retrievals over a 4×4 km$^2$ domain centred on the point source. The control observations for the SBMP and MBMP retrievals are from 2 August 2020 (Sentinel-2A) and 28 July 2020

(Sentinel-2B), and we repeat those retrievals using control observations from 11 December 2018 (Sentinel-2A) and 16 December 2018 (Sentinel-2B) to test sensitivity to the control scene.

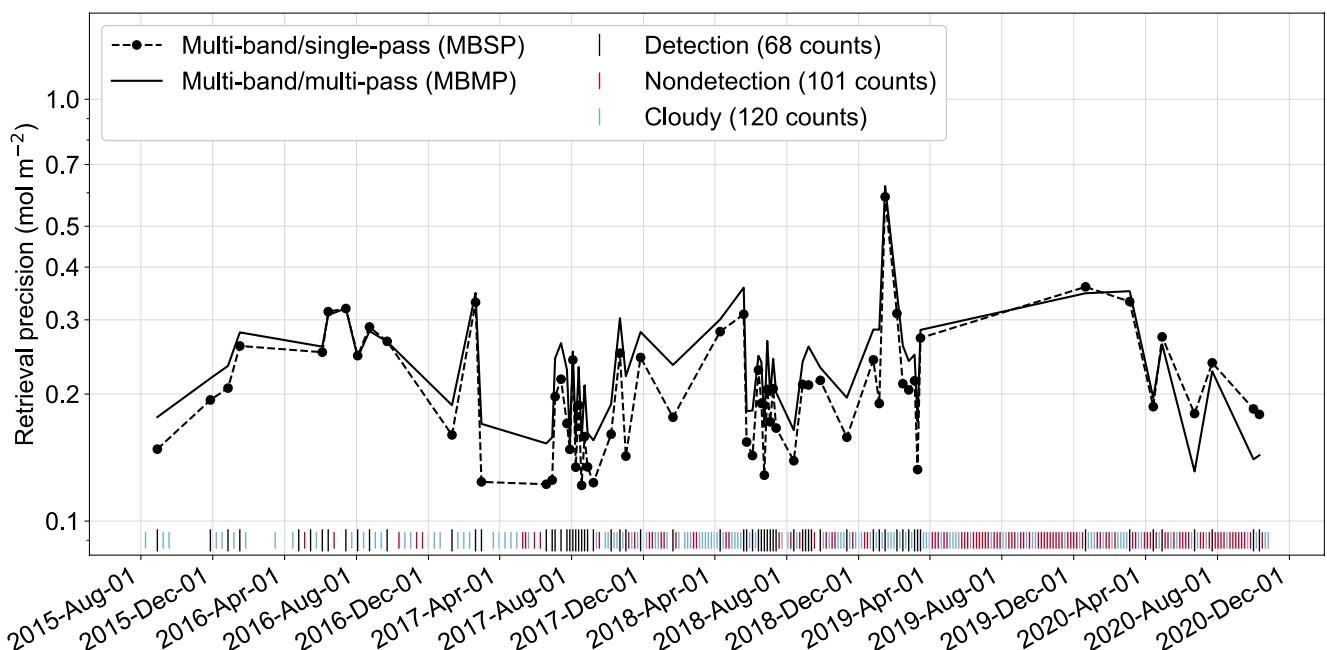

**Figure 7:** Same as Figure 5 but for the Korpezhe compressor station point source in Turkmenistan and excluding the SBMP retrieval with very coarse precision.

345          Figure 7 shows how single-pixel column retrieval precision in the Korpezhe scene varies over time. In contrast with our results for the Hassi Messaoud oil field, here we find that the MBSP retrieval performs slightly better than MBMP, with a mean column retrieval precision of 0.21 mol m$^{-2}$ (33% of background) compared to 0.24 mol m$^{-2}$ (37%) for MBMP. SBMP performs much worse (1.63 mol m$^{-2}$, 251%) and is omitted from Figure 7. This reflects a larger inter-pass variability in band-12 reflectances than for Hassi Messaoud. We obtain worse multi-pass (SBMP and MBMP) results when performing column

retrievals with our second set of control observations, from December 2018 instead of July-August 2020, with mean scene-



wide (non-plume) standard deviations of 0.39 mol m$^{-2}$ for the MBMP retrieval and 1.78 mol m$^{-2}$ for the SBMP retrieval. The poor performance of multi-pass retrieval methods could be due to variable surface conditions.

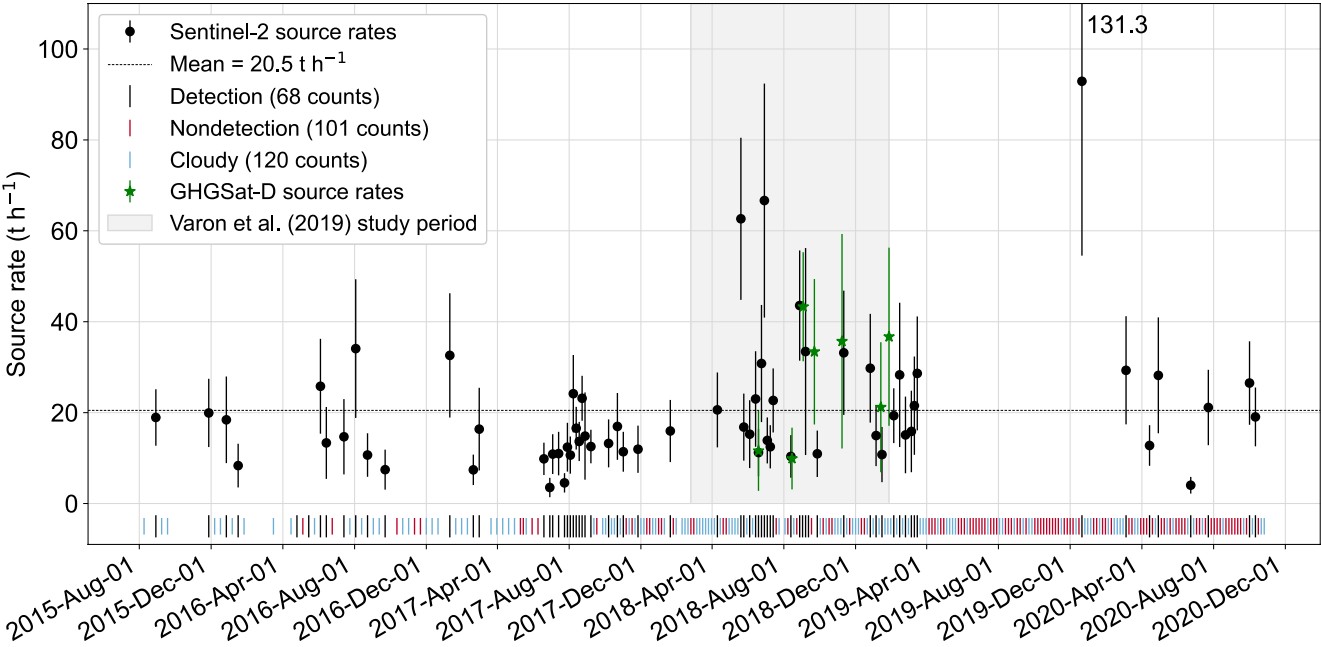

**Figure 8:** Same as Figure 6, but for the Korpezhe oil/gas field compressor station point source and using the MBSP retrieval. The green points are source rates previously estimated from GHGSat-D observations (Varon et al., 2019).

Figure 8 shows the resulting time series of source rates. We neglect 4 plume detections for which retrieval artefacts were particularly strong and indistinct from the plume, leaving a total of 64 source-rate estimates. The Korpezhe compressor station source when detected is more than twice as strong as the Hassi Messaoud oil field source, with mean emissions ± standard deviation of 20.5 ± 14.8 t h$^{-1}$ (ranging from 3.5 to 92.9 t h$^{-1}$) and extending over the full 5-year observation period. Plumes were detected in 65% of cloud-free observations from August 2015 until mid-March 2019, after which emissions apparently stopped for 8 months. Emissions then resumed in mid-December 2019 but less persistently, with plume detections in only 20% of cloud-free observations. Using the full-record 40% persistence rate and assuming no diurnal variability in emissions, we estimate total methane emissions of 373 Gg from the compressor station between 29 August 2015 and 26 October 2020, or about 75 Gg per year. This represents 5% of the annual national methane emission of 1.4 Tg from the oil/gas sector reported by the Turkmen government to the United Nations Framework Convention on Climate Change (UNFCCC; Scarpelli et al., 2020b).

Varon et al. (2019) previously inferred emissions from the Korpezhe point source using GHGSat-D observations from February 2018 to January 2019. They detected plumes on 7 of 13 clear-sky passes and these are shown in Figure 8. They reported mean emissions of 27.4 ± 12.2 t h$^{-1}$ for those plumes, in agreement with the mean Sentinel-2 estimate of 25.4 ± 16.3 t h$^{-1}$ for the same observation period and 20.5 ± 14.8 t h$^{-1}$ for the full 5-year time series. Only one day (19 June 2018) had joint





observation by GHGSat-D and Sentinel-2 (about one hour after the GHGSat-D pass). Plumes were detected on that day by both instruments with a source rate of $11.2 \pm 5.2$ t h$^{-1}$ for Sentinel-2 and $11.6 \pm 8.8$ t h$^{-1}$ for GHGSat-D. Based on these results, Sentinel-2 and GHGSat-D quantifications of methane plume columns and source rates for the Korpezhe compressor station appear to be consistent.

The long hiatus in 2019 and reduced activity in 2020 apparently reflect interventions by the facility operators to control methane emissions from the compressor station. After the GHGSat-D discovery in 2019 of anomalously large emissions from this facility, GHGSat alerted the operators to the situation through diplomatic channels (bne Intellinews, 2019). The lower frequency of plume detections from March 2019 onwards illustrates how satellite observations can help curb methane emissions, and the resumption in December 2019 shows the importance of sustained monitoring.

**5 Conclusions**

We have demonstrated the value of Sentinel-2 satellite observations for detecting and quantifying anomalous methane point sources at 20-m pixel resolution and with frequent revisit rates. Our methane retrievals use reflectance measurements from only two coarse-resolution SWIR bands (band 11, ~1560-1660 nm; and band 12, ~2090-2290 nm), but contrast between these two bands and/or with a non-plume control scene enables plume detection and quantification for sources greater than ~3 t h$^{-1}$

in scenes with favourable surface types.

We presented three different methods for retrieving methane column enhancements from the Sentinel-2 SWIR measurements in bands 11 and 12, and we applied them to monitor two point sources in the Hassi Messaoud oil/gas field of Algeria (October 2019 to August 2020, 121 passes) and in the Korpezhe oil/gas field of Turkmenistan (August 2015 to October 2020, 291 passes). The single-band/multi-pass (SBMP) retrieval relates methane column enhancements to changes in band-12

reflectances between satellite passes, with one of the passes sampling a control scene with no plume; the multi-band/single-pass (MBSP) retrieval compares band-11 and band-12 reflectances on a single pass; and the multi-band/multi-pass (MBMP) retrieval combines MBSP retrievals on different satellite passes to remove artefacts from the retrieval field. These three methods represent different approaches for simultaneously retrieving the surface albedo and methane column. We found that the MBMP retrieval generally performs best, with finer precision and lower source detection limits for most of the scene types

we examined. In the Korpezhe case, where it was difficult to define a good control observation because of variable surface conditions, the single-pass MBSP method performed slightly better.

Sentinel-2 SWIR bands 11 and 12 do not provide enough spectral information to retrieve albedo simultaneously with column concentrations of methane, water vapor, and $CO_2$, but this does not prevent mapping of strong methane plumes. Water vapour would be indistinguishable from methane without prior knowledge of the facility observed, but it is generally not co-

emitted from methane point sources and we find only a weak (6%) sensitivity of our methane retrievals to a wide range of background water vapour columns. Any $CO_2$ co-emitted with methane—for example, from active flare stacks in oil/gas fields



—would produce a low bias in the methane estimate, because band 11 contains stronger $CO_2$ absorption lines than band 12. However, flare stacks do not produce anomalously large methane emissions suitable for monitoring with Sentinel-2.

The ability of Sentinel-2 to detect methane plumes depends strongly on surface conditions. MBMP retrieval precisions

for single-pixel methane column enhancements range from 27% for the fairly homogeneous and arid Hassi Messaoud scene to more than 200% for farmland and urban mosaic scenes. For 27% precision we estimated a plume detection limit of 2.6 t h$^{-1}$. This is at the high end of methane point sources in normal operating conditions, but can detect anomalously large point sources that contribute disproportionately to emissions from a given region or sector. Sentinel-2 methane column retrievals for heterogeneous surfaces could be improved in the future with image segmentation to separately retrieve methane concentrations

over different surface types within a scene.

Application to the Hassi Messaoud and Korpezhe point sources demonstrated the value of the Sentinel-2 observations for long-term anomalous emission monitoring. The Hassi Messaoud point source was active from 9 October 2019 to 9 August 2020, and was detected in 93% of cloud-free passes during this period, with a pass every 2.5 days on average. From 101 plume detections we estimated source rates in the range 2.6-59.1 t h$^{-1}$, with a mean rate of 9.3 t h$^{-1}$ and total emissions over the 10-

month period of roughly 66 Gg. The Korpezhe point source was active from at least 29 August 2015 to 26 October 2020, with observations every 5 days after both Sentinel-2 satellites became operational. A plume was detected in 65% of cloud-free observations leading up to March 2019, with source rates in the range 3.5-92.9 t h$^{-1}$ (averaging 20.5 t h$^{-1}$), consistent with previously-reported observations by the GHGSat-D satellite instrument from February 2018 to January 2019 (Varon et al., 2019). We see in the Sentinel-2 data that emissions shut down in March 2019 following communication by GHGSat with the

facility operators, but then resumed in December 2019. The total emission from the Korpezhe point source over the 5-year period of Sentinel-2 data is estimated to be 373 Gg.

Our demonstration of the Sentinel-2 capability for high-frequency monitoring of anomalously large methane point sources can be readily extended to other multispectral satellite instruments with similar SWIR spectral bands, including Landsat 7 and Landsat 8. In the future, these satellite observing systems can function as early alert systems for identifying

anomalously large emissions at industrial facilities, enabling prompt corrective action and significant abatement of total methane emission on regional/national scales. Multispectral satellite data will be particularly effective when combined with targeting instruments like the GHGSat instrument suite for finer-resolution detection, and from global mapping instruments like TROPOMI that can place these point source emissions in a regional context.

**Data availability**

All Sentinel-2 satellite data used for this study are publicly available through the Copernicus Open Access Hub (scihub.copernicus.eu/). The GEOS-FP wind data are publicly available through the NASA Climate Data Services portal (nccs.nasa.gov/services/climate-data-services). The HITRAN line spectra are publicly available through the HITRANonline database (https://hitran.org/).





**Author contribution**

D.J.V., D.J., J.M., I.S., and D.J.J. contributed to study conceptualization. D.J.V., D.J., J.M., I.S., and D.G. contributed to methods development and data analysis. D.J.V. wrote the original draft and all authors reviewed and edited the manuscript.

**Competing interests**

The authors declare that they have no conflict of interest.

**Acknowledgements**

DJJ was supported by the NASA Carbon Monitoring System.



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
