# Peer review of "High-frequency monitoring of anomalous methane point sources with multispectral Sentinel-2 satellite observations"

_Atmospheric Measurement Techniques, 2020_

## Referee Comment (RC1) · Edward Malina (Referee) · 14 Jan 2021

Please see attached review.

Please also note the supplement to this comment:
https://amt.copernicus.org/preprints/amt-2020-477/amt-2020-477-RC1-supplement.pdf

---

## Referee Comment (RC2) · Anonymous Referee #2 · 22 Jan 2021

In the manuscript "High-frequency monitoring of anomalous methane point sources with multispectral Sentinal-2 satellite observations" Daniel J. Varon and colleagues investigate the potential of observations from the Sentinel-2 MultiSpectral satellite Instrument (MSI) to identify and quantify emissions from strong $CH_4$ point sources over favourable surfaces. They present three different approaches to analyse integrated radiances from two different measurement bands (around 1650 nm and 2200 nm) in terms of atmospheric $CH_4$ concentrations, which are then converted to emissions. The algorithms are eventually applied to observations near a well pad in an oil field in Algeria and near a compressor station in an oil/gas field in Turkmenistan. In addition, the results are compared to observations and emissions from a second satellite, the GHGSat-D demonstration satellite instrument.

The introduced approach has the potential to fill a gap in the current observing system for $CH_4$ emissions. Although it is only applicable to large emissions (> 3 t h$^{-1}$) occurring over quasi-homogeneous surfaces, it utilizes observations from satellites, so far not used for greenhouse gas retrievals, having fine spatial resolution and a good re-visiting time. The manuscript fits well in the scope of AMT and I recommend publication after some modifications along the line of the comments below.

The manuscript is well-written and conclusive. The methods are described in a comprehensive way, although the authors should elaborate a bit more detailed on their basic assumptions regarding surface reflectance and scattering effects in both bands. In addition, further sensitivity tests for the inferring gas $CO_2$ (as already done for water vapor) would be appropriate. See specific comments below.

Specific comments:

**P4, L92f**: How would the mean optical depth of $CO_2$ in both bands compare to that of $CH_4$?

**P4, L103**: Concerning your assumption about the surface and aerosol reflectance properties in bands 11 and 12 – could you provide any references for your assumption that they are similar in both bands? According to a publication from Chen et al. (2006) using VIRS and MODIS observations (their Table 1), it seems that depending on surface type, the e.g., surface reflectance properties can differ at 1620 nm and 2130 nm.

Would you have any possibility using model simulations to investigate the effect of variable surface and aerosol reflectance properties on your retrieval (I assume the statement "The model accounts… not for … scattering" in L106f refers only to aerosol scattering)? I could also imagine that these effects are partly captured by your factor $c$ in Eq. 1.

**P5, L115f**: I agree with your statement that "Aerosol effects… (1) uniform across the scene, (2) … not co-emitted with methane…", however, what happens if the 'general' atmospheric aerosol load across the scene is relatively heavy on the day of the control observation but relatively clean on measurement days, or vice versa? Would this have an influence on the retrieved $CH_4$ columns or also be correctly captured by $c$?

**P5, L137**: Any ideas why the fractional signals from the satellites are different? Is it related to the different windows and/or to slightly different instrument characteristics / calibration?

**P5, L141f**: Similar tests as done for water vapour could potentially also be done for $CO_2$ to quantify any induced bias on the $CH_4$ column as indicated in L402.
Additionally, would a relatively high atmospheric $CO_2$ background concentration in the observed scene during e.g., the control observation, have a large influence on the retrieved $CH_4$ column if the

observation itself only exhibit low atmospheric $CO_2$ background concentrations (in the case of the multi-band approaches)?

**P5, L142**: "… this affects $m_{SBMP}$ by only 6%...": I do not quite understand what exactly the 6% refers to. Does it mean, a potential change in water vapour would cause a change of $m_{SBMP}$ by 6%, which would then be larger than the ~3% caused by a doubling of $CH_4$ (L137), OR is the ~3% change caused by $CH_4$ only modified by the stated 6% from water vapour so that, in this special case, the overall effect of water vapour on $m_{SBMP}$ would be only around 0.18% (0.06*0.03)?

**P8, L199f**: Would it be possible to also add figures of top-of-atmosphere reflectances for bands 11 and 12 for the scenes Korpezhe, Savannah, Urban and Farmland? That would be a good opportunity to visualize the variability in surface reflectivity for the scenes as mentioned in paragraph L223-228 (although I understand that the manuscript is already quite figure-heavy).

**P12, L309**: I would suggest also adding the median value for MBMP as done for SBMP and MBSP in L313.

**P13, L314**: I think "less" would fit better here: "…because it is not subject to…" → …because it is less subject to…

**P13, L318f**: As indicated at the beginning of Sect. 4.2, depending on the scene, one or the other method may work better. How would Fig. 6 and the mean emission look like if SBMP and MBSP were used instead of MBMP? Would one get a bias, enlarge the error bars or would the single emissions get 'more' variable? Additionally, how does a change of the control scene influence the retrieval precision (Fig. 5)?

**P13, L323f**: Have you, in addition to the plume free scenes before and after the mission, also tested the 8 plume free scenes within the mission for the Hassi Messaoud oil field or are there any reasons why they cannot be used as control observations? Would their RMS value also be around 20%?

**Fig. 1**: I would suggest adding column to "slant optical depths" → slant column optical depths

**Fig. 2**: Just a comment: The SBMP retrieval appears to have some issues with surface elevation (or shadows caused by hills due to different illumination conditions on 2019-Oct-06 and 2019-Nov-20) if I interpret the red areas in the upper left and lower right corner in (d) correctly. Otherwise, the remaining part of that plot looks 'cleaner' than for the MBSP retrieval in (e).

Technical corrections:

**P4, L87**: I would suggest capitalizing "high-resolution transmission molecular absorption (HITRAN)" → HIgh-resolution TRANSmission molecular absorption (HITRAN) database

**P18, L444**: "DJJ" → D.J.J.

**Fig. 4**: Please add labels (a) and (b) to the two subfigures.

References

Chen, Y., Sun-Mack, S., Arduini, R. F., and Minnis, P.: Clear-sky and surface narrow band albedo variations derived from VIRS and MODIS Data, in: conference on cloud physics, CD ROM edition, 5.6 Atmospheric radiation 12th, Conference, Atmospheric radiation, vol. 12, Atmospheric radiation, Boston, Mass., USA, 2006.

---

## Author Comment (AC1) · 2 Mar 2021

**Responses to Reviewer #1**

We thank the reviewer for their comments and questions. Our responses are formatted as follows:

*The reviewer's comment/question (numbered) is written in black italic text.*

> Our responses are written in normal black text (indented).

> The revised text as it appears in the manuscript is written in normal blue text (indented), with relevant changes underlined.

Line numbers refer to the edited manuscript. We have also provided a tracked-changes document, but that has different line numbers.

*1. There is no mention of instrument noise in the paper, for example equation (2) (and the related equations) rely on a fractional absorption model to infer column enhancements from reflectance changes in the measured radiance. The measured radiances will include instrument noise, with R and R' not showing the same noise due to capture under difference conditions. The fractional absorption model does not seem to account for instrument noise, meaning that the minimisation step between the R and m values may be underestimated. It is possible that the instrument noise is accounted for in the forward model calculations as identified in Jervis et al. (2020), if this is so, it should be explicitly stated, and the noise model identified.*

Thank you for raising this point. An instrument noise model does not need to be specified to perform the methane retrieval. However, for the purpose of assessing retrieval precision, we assume that the instrument noise is normally distributed and uncorrelated. We now state this in the text on L. 203-206.

We estimate single-pixel column retrieval precision as the standard deviation of non-plume methane enhancements across the scene, assuming uncorrelated and normally-distributed instrument noise. We obtain precisions of 0.32 mol m$^{-2}$ (roughly 49% of background) for the SBMP retrieval and 0.31 mol m$^{-2}$ (roughly 48% of background) for the MBSP retrieval.

*2. Further to this, do Sentinels-2A and B exhibit different behaviour due to differing instruments, and different noise levels?*

We now explain that the fractional signal change is different between the two instruments due to their different spectral ranges (L. 153-154).

The difference between these two values is due to the instruments' slightly different spectral ranges (Figure 1).

*3. The paper would benefit from some more validation, understandably this is challenging due to minimal measurements at this resolution (both spatially and temporally), and while there are some comparisons with GHGSat-D in this study (which show good results), I would have liked to have seen some TROPOMI retrievals over the test cases. These comparisons would distinctly show how TROPOMI cannot identify these plumes (or maybe they can), while Sentinel-2 can.*

We explain in the introduction that TROPOMI spatial resolution is too coarse to resolve individual point sources except in special cases (e.g., Pandey et al., 2019), and have demonstrated this in a previous publication (Varon et al., 2019). We now direct the reader to this previous work in our discussion of TROPOMI point-source detection capabilities (L. 37-38).

The TROPOspheric Monitoring Instrument (TROPOMI) aboard the Sentinel-5 Precursor satellite provides daily global methane measurements at up to 5.5×7 km$^2$ pixel resolution (Hu et al., 2018; Schneising et al., 2019), sufficient to detect major accidental blowouts at oil/gas facilities (Pandey et al., 2019), but generally too coarse to resolve point sources

(Varon et al., 2019), which are often spatially clustered and typically produce plumes <1 km in scale (Frankenberg et al., 2016; Duren et al., 2019).

*4. The authors briefly state at the beginning of page 10 that only unusually strong methane sources are detectable using the methods identified in this paper (~>3 t /hr). However, there is very little detail in the paper about how common such sources are and what they might be. The statement that most large sources fall between 1-10 t/hr is difficult to conceptualise without context. Therefore the overall utility of this method with Sentinel-2 remains unclear, and this should be improved. I recommend including a section placing the sources in the case studies in this paper in context with other global sources e.g. biomass burning events, or coal mining emissions, both significant methane sources (Saunois et al., 2020), therefore giving readers an idea of how the work in this paper could be applied globally.*

    Thank you for raising this important point. We elaborate on the significance of methane point sources of this magnitude (L. 266-272).

    The largest methane point sources under normal operating conditions (landfills, wastewater treatment plants, and the vents of underground coal mines) can emit 1-10 t h$^{-1}$ as annual means (Jacob et al., 2016; Varon et al., 2020; Scarpelli et al., 2020a). Our detection-limit estimates for Sentinel-2 thus restrict application to unusually strong sources, but such sources are routinely detected in abnormal operations (Pandey et al., 2019; Varon et al., 2019; Cusworth et al., 2020). Duren et al. (2019) found that sources stronger than 2.5 t h$^{-1}$ accounted for <5% of point source emissions in California, but Frankenberg et al. (2016) found that they were responsible for more than 25% in the Four Corners oil/gas production region of New Mexico.

*5. P2 L 36 - (Schneising et al., 2020) should be identified here as well.*

    Schneising et al. (2020) investigate methane emissions from oil/gas basins using TROPOMI observations. This is not directly relevant to the passage in question, which is about point sources, but we realize that the work of Schneising et al. (2019) on TROPOMI methane retrievals should be acknowledged here (in addition to the Hu et al. study). Thank you for bringing this to our attention. We have added the Schneising et al. (2019) citation to L. 36 and to the reference list.

    The TROPOspheric Monitoring Instrument (TROPOMI) aboard the Sentinel-5 Precursor satellite provides daily global methane measurements at up to 5.5×7 km$^2$ pixel resolution (Hu et al., 2018; Schneising et al., 2019)…

*6. P2 L45 – It'd be useful to identify the main aims of Sentinel-2 in this section i.e. identifying land changes etc. This would help differentiate the scope of this study from the main aims of the Sentinel-2 mission.*

    Thanks for this suggestion. We now briefly discuss the original aims of the Sentinel-2 mission in the introduction (L. 47-49).

Here we demonstrate the capability of the current Sentinel-2 twin satellites to detect and quantify strong methane point sources globally with both fine pixel resolution and frequent revisits. Sentinel-2 was originally designed to provide operational data products for environmental risk management, land cover classification, land change detection, and terrestrial mapping, as a complement to the Landsat and SPOT satellite missions. It comprises two satellites positioned 180° out of phase in the same sun-synchronous orbit, with an equator-crossing time of 10:30 (local solar time) at the descending node. Sentinel-2A (S2A) was launched in June 2015 and Sentinel-2B (S2B) in March 2017.

*7. P2 L53 – While 2300 nm does include significantly more methane spectral lines than 1600 nm, solar irradiance is several times higher at 1600 nm, and surface reflectance is typically higher at 1600 nm. Therefore, one has to be careful about interpreting the next statement '2300 nm being considerably more sensitive to methane than 1600 nm'. I recommend this section be qualified with some statement about solar irradiance.*

Thank you for raising this point. Differences in solar irradiance between bands are not a concern because Sentinel-2 bands 11 and 12 have identical SNR of 100 at their respective reference radiances. Therefore we are only concerned with reflectance differences between the bands. If the reflectance in a given band is systematically higher than in another, this is accounted for by our factor 'c'. We clarify in the text on L. 57-59.

Band 12, overlapping with the stronger and broader 2300-nm feature, is considerably more sensitive to methane than band 11. This is despite the comparatively lower solar irradiance in band 12, because the two bands have equivalent signal-to-noise ratios (SNR) at their respective reference radiances (Drusch et al., 2012).

*8. P2 L54 – "Band 11 can therefore be used as a proxy for the continuum". It is not clear what is meant here? Surface reflectance values can still be significantly different between Band 11 and 12, especially in high albedo scenes. This combined with higher solar irradiance in band 11, indicate that there are significant differences between the bands.*

Please see our response to comment #7.

*9. P3 L67 – Are there any upcoming instruments that this method could be applied to?*

We add that our methods can be applied to the Sentinel-3 SLSTR instrument as well.

Our techniques are developed with a focus on Sentinel-2 satellite observations, owing to the exceptional spatial and temporal resolution of MSI data, but can easily be extended to observations from other multispectral surface imagers with similar spectral bands, such as the Landsat 7 and Landsat 8 instruments with 30-m pixel resolution and a combined 8-day revisit rate, or the Sentinel-3 SLSTR instrument with 500-m pixel resolution and daily revisits.

*10. P4 L93 – The risk of artefacts due to water vapour lines is identified here, but artefacts due to spectroscopic database uncertainties have not been identified. Since 2300 nm is very complex*

*spectrally, it is likely retrieval artefacts will exist, especially over high reflectance environments where the lines can saturate. Therefore, I recommend to include a short discussion here (or elsewhere as appropriate) on the potential impact of spectroscopic parameter uncertainty.*

Thank you for this suggestion. We now note the possibility of quantification errors from database uncertainty (L. 104-105).

Additional quantification errors may arise from spectroscopic database uncertainties.

*11. P4 L100 – Here the concept of residual radiance analysis is introduced, the core of the analysis of this paper. The sentence reads a little bit as though this is a new technique, which it is not. I recommend that a discussion or identification of past uses of methane detection with residual radiance be included here e.g. (Leifer et al., 2006; Roberts et al., 2010).*

Thank you for making this important point. We now mention these previous studies and another one by Innocenti et al. (2017) that used similar techniques (L. 115-117).

But one can also in principle retrieve methane column concentrations together with surface albedo from just two spectral measurements, one featuring methane absorption and one not. This could be done for a single spectral band by comparing observations of the same scene with and without a methane plume. It could also be done for a single scene with two adjacent spectral bands that are sufficiently close to have similar surface and aerosol reflectance properties, but differ in their methane absorption properties. Similar techniques have previously been used to retrieve methane column concentrations from ground-based (Innocenti et al., 2017) and airborne (Leifer et al., 2006; Roberts et al., 2010) remote sensing instruments. We demonstrate this here using Sentinel-2 bands 11 and 12.

*12. P4 L105 – Here a radiative transfer model is briefly described, can access to this model be provided? Fundamentally, the results shown in the paper need to be reproducible, and the RTM is key in this regard.*

We will make the radiative transfer model code available upon request and have added a Code Availability section stating this (L. 469-470).

*13. Further, no mention of the model used for surface reflectance is identified which is surprising given the importance of reflectance in this study. Assuming the same model used in Jervis et al. (2020) is employed, is a Lambertian model sufficient at such high spatial resolution? Would not a BRDF model yield improved results, possibly dealing with some of the heterogeneous scenes?*

We agree with the reviewer that the surface reflectance model should be explicitly mentioned. This information has been added to L. 125. The scaling factor $c$ will automatically account for any angle-dependent reflectance effects in the SBMP retrieval method. For the MBSP method, the reflectance measurements of different bands are at

the same observation and thus scattering angle. For these reasons, we don't believe that incorporating a BRDF reflectance model will improve results.

The incident solar irradiance is from Clough et al. (2005) and we assume the surface reflectance to be Lambertian.

*14. P4 L110 – Presumably the HITRAN2016 database?*

Fixed, thanks.

*15. P5 L115 – The assumption about aerosols is not well justified here, especially given the location of the case studies in this paper are in desert regions, well known to be affected by aerosols.*

Aerosols in desert scenes do not contradict the justifications we provide, but to clarify we have rephrased and expanded on this justification (L. 128-131).

Aerosol effects are ignored because methane sources generally do not co-emit aerosols, and because background aerosol such as from dust can be assumed uniform across a given scene (like water vapour and $CO_2$). Neglecting aerosol scattering may produce methane retrieval errors of a few percent (Huang et al., 2020), but as we show below this is much smaller than typical Sentinel-2 methane retrieval errors.

*16. P5 Equation 2 – It is unclear as to how instrument noise in band 12 is accounted for in this calculation. The radiances used in equation 1 will include instrument noise, so there should be some accounting for this in equation 2?*

Please see our response to comment #1.

*17. P5 L136 – Are the differences between Sentinel-2A and 2B purely due to the spectral range differences?*

Please see our response to comment #2.

*18. P5 L141-142 – It is very unclear what the authors have undertaken here with assessing the impact of water vapour. It is stated that variations in background water vapour have a minor impact of 6% (although how does a 6% variation affect the precision). But what background is being used in this test? Are the water vapour variations the max that could be seen in the US standard atmosphere, or based off the max in a tropical scene?*

The range of water vapor column abundances used reflects the range observed by Nelson et al. (2016) over land using OCO-2. We now state this explicitly (L. 158-160).

Water vapour columns over land may vary from 1 to 40 kg m$^{-2}$, as observed by the Orbiting Carbon Observatory 2 (OCO-2; Nelson et al., 2016), but under nominal observing conditions $m_{SBMP}$ varies by only 6% over this range (i.e., by roughly $\pm$ 0.002).

*19. P6 L149 – I'm not convinced that relying on similar radiance levels between the spectral bands works as suggested in this section. Even if surface reflectances are similar (which they may not be), solar irradiance and instrument noise will likely mean different SNRs between band-12 and band-11, yielding large magnitude differences. Can this all be accounted for in the 'c' factor in equation 3?*

We discuss in Section 3.5 the consequences of surface reflectance differences between bands; if the differences are too large, the retrieval becomes too noisy to resolve all but the most anomalous plumes. We address the question of solar irradiance and SNR differences between bands in our response to comment #7. Our retrieval demonstration in Section 3.4 and our documentation of precision levels in Figures 3, 5, and 7 show that the 'c' factor can indeed account for broad differences between spectral bands and satellite passes.

*20. P8 L197 – It is stated that 'c' is used to account for scene-averaged differences in reflectances between satellite passes or bands. However earlier in the text (P6 L150), 'c' is identified as being used to account for calibration differences, implying minor variations. This line should be moved up to p6 to give more detail about the use of 'c'.*

The parameter 'c' can be interpreted differently in the SBMP and MBSP methods. In the former, it accounts for scene-wide differences between passes; in the latter, scene-wide differences between bands. We explain on L. 170-172 that in the MBSP method 'c' addresses both calibration differences *and* spectral dependences of the albedo. For clarity we now also elaborate on the meaning of 'c' in the SBMP method (L. 143-144; see our response to Reviewer #2 comment #4). Our statement that 'c' accounts "for scene-averaged differences in reflectance between satellite passes or bands" (L. 219-221) covers both the SBMP and MBSP methods.

*21. P9 L218 – The term 'plume detection limit' is not explicitly identified, please explain this term and how it is calculated. This term is also used in Figure 3, with no explanation as to what this is or how it is calculated.*

We explain how these values are obtained in Section 3.5 (L. 261-266) and in the caption of Figure 3. We have rephrased our explanation for clarity.

The single-pixel column retrieval precisions of Figure 3 can be related to empirical plume detection limits—the smallest source rate Sentinel-2 can detect in a given scene—by using the long-term observation records for the Hassi Messaoud and Korpezhe scenes shown in Section 4. For Hassi Messaoud with 27% retrieval precision, the minimum detected source is 2.6 t h$^{-1}$. For Korpezhe with 33% retrieval precision but more temporally variable surface conditions, the minimum detected source is 3.5 t h$^{-1}$. The plume detection limit is expected to be proportional to precision (Jacob et al., 2016) and is shown in Figure 3 for the savannah, urban, and farmland scenes by scaling of the Hassi Messaoud MBMP value.

*22. P9 L227 – This MBMP precision is still significantly worse than the homogeneous scenes, is this still due to reflectance errors?*

> Yes, and also because this surface is darker than the Algerian and Turkmen scenes. For example, comparing to the Hassi Messaoud scene, the farmland scene is 4 times darker in band 11 and 9 times darker in band 12 (mean reflectances on L. 239-241).

*23. P10 L221 – This section should go in the conclusions.*

> We assume the reviewer is referring to the following passage, beginning around L. 241: "Our detection-limit estimates for Sentinel-2 thus restrict application to unusually strong sources, but such sources are frequently detected in abnormal operations (Pandey et al., 2019; Varon et al., 2019; Cusworth et al., 2020)." We now reiterate this point in the conclusions (L. 441-443).

> For 27% precision we estimated a plume detection limit of 2.6 t h$^{-1}$. This is at the high end of methane point sources in normal operating conditions, but can detect anomalously large point sources that contribute disproportionately to emissions from a given region or sector and have been routinely observed by satellite (Pandey et al., 2019; Varon et al., 2019; Cusworth et al., 2020).

*24. P10 L253 – With regards to TROPOMI, it'd be useful to contrast the results of TROPOMI with those found by the methods shown in this paper. Therefore providing direct proof of the utility of this method with Sentinel 2, if such data is available at this time.*

> Please see our response to comment #3.

*25. P2 L35 – TROPOMI now operates at a spatial resolution of 3.5x5.5 km2.*

> That is true for e.g., the $NO_2$ retrieval, but not for methane. The nominal resolution of the TROPOMI methane product is 5.5x7 km$^2$.

*26. P4 L88 – HITRAN -> HITRAN2016*

> Fixed, thanks.

---

## Author Comment (AC2) · 2 Mar 2021

**Responses to Reviewer #2**

We thank the reviewer for their comments and questions. Our responses are formatted as follows:

*The reviewer's comment/question (numbered) is written in black italic text.*

Our responses are written in normal black text (indented).

The revised text as it appears in the manuscript is written in normal blue text (indented), with relevant changes underlined.

Line numbers refer to the edited manuscript. We have also provided a tracked-changes document, but that has different line numbers.

1. P4, L92f: How would the mean optical depth of CO2 in both bands compare to that of CH4?

We have added the requested comparison to the text (L. 100).

The mean methane optical depth in band 12 is five times larger than that in band 11. The mean  $CO_2$  optical depth is about 5 times larger in band 11 and 24 times smaller in band 12 than that of methane.

2. P4, L103: Concerning your assumption about the surface and aerosol reflectance properties in bands 11 and 12 – could you provide any references for your assumption that they are similar in both bands? According to a publication from Chen et al. (2006) using VIRS and MODIS observations (their Table 1), it seems that depending on surface type, the e.g., surface reflectance properties can differ at 1620 nm and 2130 nm.

The reviewer is right that similarity in spectral reflectance between the two bands is not guaranteed. Our multiband methods (MBSP and MBMP) will only work well when reflectances in the two bands are indeed similar. We explain this on lines 212-215:

"One might expect the MBMP retrieval to be strictly superior to the SBMP and MBSP retrievals since it exploits both multi-band and multi-pass information to derive methane column concentrations. However, this may not be the case for scenes with large differences in surface reflectance between bands 11 and 12 due to strong spectral dependence of the albedo (in which case SBMP might be superior), or if no good reference observation with consistent surface reflectance is available (in which case MBSP might be superior). We investigate the dependence of the methane retrieval precision on surface type and retrieval method in the following section."

3. Would you have any possibility using model simulations to investigate the effect of variable surface and aerosol reflectance properties on your retrieval (I assume the statement "The model accounts ... not for ... scattering" in L106f refers only to aerosol scattering)? I could also imagine that these effects are partly captured by your factor c in Eq. 1.

Indeed, scene-wide differences in surface and aerosol reflectances between spectral bands and satellite passes are captured by the factor c, but variability in reflectances across the scene leads to errors. We explain that in the text on lines 219-221:

"Our Sentinel-2 methane column retrievals require close agreement between MSI surface reflectances measured on different satellite passes and/or in different SWIR spectral bands to isolate methane plumes in a scene. The scaling factor *c* of Equations (1) and (3) is fitted to account for scene-averaged differences in reflectance between satellite passes or bands, but inter-pixel variability in surface reflectance remains and will be a source of error."

A detailed study of the impacts of surface/aerosol reflectance heterogeneity on Sentinel-2 methane retrievals over different scenes would be valuable, but is beyond the scope of the

present work. We discuss generally the question of surface properties in Section 3.5, showing in equation (6) the relationship between reflectance variability and methane retrieval error.

4. P5, L115f: I agree with your statement that "Aerosol effects... (1) uniform across the scene, (2) ... not co-emitted with methane...", however, what happens if the 'general' atmospheric aerosol load across the scene is relatively heavy on the day of the control observation but relatively clean on measurement days, or vice versa? Would this have an influence on the retrieved CH4 columns or also be correctly captured by c?

If the aerosol effect is uniform across the scene, then it will be captured by c. To clarify, we elaborate on the meaning of c in the SBMP method (L. 143-144). Please also see our response to comment #15 by reviewer #1.

... where c is a scaling factor that adjusts for scene-wide changes in brightness between satellite passes. Such changes could be due to variable observation zenith angles, atmospheric conditions, or surface conditions.

**5. P5, L137: Any ideas why the fractional signals from the satellites are different? Is it related to the different windows and/or to slightly different instrument characteristics / calibration?**

We clarify that the difference in signal change is due to slightly different band positions/widths (L. 153-154).

The difference between these two values is due to the instruments' slightly different spectral ranges (Figure 1).

6. P5, L141f: Similar tests as done for water vapour could potentially also be done for CO2 to quantify any induced bias on the CH4 column as indicated in L402.

 $CO_2$  variability is much lower than that of H2O. We performed the same test as for H2O, but varying CO2 between 400 and 410 ppm. This affects mSBMP by 0.01% or less (compared with 6% for the wide range of water vapor levels tested). We explain this on L. 162-163.

Water vapor columns over land may vary from 1 to 40 kg m-2, as observed by the Orbiting Carbon Observatory (OCO-2; Nelson et al., 2016), but under nominal observing conditions  $m_{\text{SBMP}}$  varies by only 6% over this range (i.e., by roughly  $\pm$  0.002). Variability in background CO2 is much less than for water vapour and has virtually no effect on  $m_{\text{SBMP}}$ ; varying the CO2 column between 400 and 410 ppm changes  $m_{\text{SBMP}}$  by  $\leq$  0.01%.

7. Additionally, would a relatively high atmospheric CO2 background concentration in the observed scene during e.g., the control observation, have a large influence on the retrieved CH4 column if the observation itself only exhibit low atmospheric CO2 background concentrations (in the case of the multi-band approaches)?

Please see our response to comment #6.

8. P5, L142: "... this affects mSBMP by only 6%...": I do not quite understand what exactly the 6% refers to. Does it mean, a potential change in water vapour would cause a change of mSBMP by 6%, which would then be larger than the  $\sim$ 3% caused by a doubling of CH4 (L137), OR is the  $\sim$ 3% change caused by CH4 only modified by the stated 6% from water vapour so that, in this special case, the overall effect of water vapour on mSBMP would be only around 0.18% (0.06\*0.03)?

Thank you for raising this important question. We need to clarify that the 6% change is expressed relative to mSBMP, which is itself a relative quantity. So indeed, it is 6% of a roughly 3% fractional signal change, or a difference of about 0.2% fractional signal change, as the reviewer describes. We explain this in the text on L. 160.

Water vapor columns over land may vary from 1 to 40 kg m-2, as observed by the Orbiting Carbon Observatory (OCO-2; Nelson et al., 2016), but under nominal observing conditions  $m_{\text{SBMP}}$  varies by only 6% over this range (i.e., by roughly  $\pm$  0.002).

9. P8, L199f: Would it be possible to also add figures of top-of-atmosphere reflectances for bands 11 and 12 for the scenes Korpezhe, Savannah, Urban and Farmland? That would be a good opportunity to visualize the variability in surface reflectivity for the scenes as mentioned in paragraph L223-228 (although I understand that the manuscript is already quite figure-heavy).

Thank you for this suggestion. The discussion of the additional scenes is intended only to illustrate the importance of surface type for the ability of Sentinel-2 to resolve methane plumes. Figure 3 and the discussion in Section 3.5 seem sufficient for this, and we agree that the manuscript already has a large number of figures. We hope that the geographic information we provide for the scenes will enable the interested reader to inspect them more closely online.

10. P12, L309: I would suggest also adding the median value for MBMP as done for SBMP and MBSP in L313.

We have added the median value (L. 339).

Precision fluctuates between about 0.1 and 0.25 mol m-2, with a mean of 0.18 mol m-2 (27% of background, value reported in Figure 3), a median of 0.16 mol m-2, and three outliers higher than 0.3 mol m-2.

11. P13, L314: I think "less" would fit better here: "...because it is not subject to ..."  $\rightarrow$  ...because it is less subject to ...

Good point. We clarify that the MBSP method is not subject to temporal variability (L. 345).

The MBSP shows the most consistent (but coarse) precision across the time series, presumably because it is not subject to temporal variability in surface conditions.

12. P13, L318f: As indicated at the beginning of Sect. 4.2, depending on the scene, one or the other method may work better. How would Fig. 6 and the mean emission look like if SBMP and MBSP were used instead of MBMP? Would one get a bias, enlarge the error bars or would the single emissions get 'more' variable?

The figure looks about the same, but with larger error bars for the MBSP and SBMP methods, and slightly different mean emission rates. We now state in the text the difference in mean values (L. 355).

Figure 6 shows the resulting time series of source rates, which range from 2.6 to 59.1 t h-1, with a mean  $\pm$  standard deviation of 9.3  $\pm$  5.5 t h-1. Mean values estimated with the SBMP and MBSP methods are within 22% of this estimate.

*13. Additionally, how does a change of the control scene influence the retrieval precision (Fig. 5)?*

We now compare the mean retrieval precision for the two multi-pass methods when using the second set of reference/control images (L. 345-347).

The SBMP and MBSP retrievals show much coarser precision, with mean (median) scene-wide standard deviations of 0.35 (0.28) and 0.38 (0.36) mol m-2, respectively (omitting the extreme outlier in October 2019 from the SBMP calculation). The MBSP shows the most consistent (but coarse) precision across the time series, presumably because it is not subject to temporal variability in surface conditions. We obtain similar multi-pass retrieval precisions using our second set of reference images, from September 2020; the mean precision is 0.39 mol m-2 for the SBMP method and 0.18 mol m-2 for the MBMP method.

14. P13, L323f: Have you, in addition to the plume free scenes before and after the mission, also tested the 8 plume free scenes within the mission for the Hassi Messaoud oil field or are there any reasons why they cannot be used as control observations? Would their RMS value also be around 20%?

We have not tested this. A good reason not to use them is that they may contain active emissions that are simply below our detection threshold. We now explain this on L. 326.

Of the 20 non-detections, 12 were due to cloud cover and 8 showed no detectable plume, indicating a plume persistence rate of 93% for cloud-free observations. Non-detections can be due source inactivity or to emission rates below the Sentinel-2 detection threshold.

15. Fig. 1: I would suggest adding column to "slant optical depths"  $\rightarrow$  slant column optical depths

Done.

16. Fig. 2: Just a comment: The SBMP retrieval appears to have some issues with surface elevation (or shadows caused by hills due to different illumination conditions on 2019-Oct-06 and 2019-Nov-20) if I interpret the red areas in the upper left and lower right corner in (d) correctly. Otherwise, the remaining part of that plot looks 'cleaner' than for the MBSP retrieval in (e).

Good point, we have taken note of this.

17. P4, L87: I would suggest capitalizing "high-resolution transmission molecular absorption (HITRAN)"  $\rightarrow$  HIgh-resolution TRANSmission molecular absorption (HITRAN) database

Good suggestion.

18. P18, L444: "DJJ"  $\rightarrow$  D.J.J.

Fixed. Thanks!

19. Fig. 4: Please add labels (a) and (b) to the two subfigures.

Done. Figure caption updated accordingly.